# Recent Trends in Nanomedicine-Based Strategies to Overcome Multidrug Resistance in Tumors

**DOI:** 10.3390/cancers14174123

**Published:** 2022-08-26

**Authors:** Muhammad Muzamil Khan, Vladimir P. Torchilin

**Affiliations:** 1Faculty of Pharmacy, The Islamia University of Bahawalpur, Bahawalpur 63100, Pakistan; 2Center of Pharmaceutical Biotechnology and Nanomedicines, Department of Chemical Engineering, Northeastern University Boston, Boston, MA 02115, USA

**Keywords:** multidrug resistance (MDR), nanomedicines, siRNA, mRNA, chemosensitizers

## Abstract

**Simple Summary:**

Cancer chemotherapy results in poor treatment response due to increased tumor resistance. There is a need to develop novel nanoparticle-based strategies to overcome resistance and improve therapeutic response. Nanomedicine-based strategies include the nanoparticle-/liposome-based delivery of drugs and siRNA to downregulate the genes causing resistance and the use of natural substances such as chemosensitizers to improve response to therapy.

**Abstract:**

Cancer is the leading cause of economic and health burden worldwide. The commonly used approaches for the treatment of cancer are chemotherapy, radiotherapy, and surgery. Chemotherapy frequently results in undesirable side effects, and cancer cells may develop resistance. Combating drug resistance is a challenging task in cancer treatment. Drug resistance may be intrinsic or acquired and can be due to genetic factors, growth factors, the increased efflux of drugs, DNA repair, and the metabolism of xenobiotics. The strategies used to combat drug resistance include the nanomedicine-based targeted delivery of drugs and genes using different nanocarriers such as gold nanoparticles, peptide-modified nanoparticles, as well as biomimetic and responsive nanoparticles that help to deliver payload at targeted tumor sites and overcome resistance. Gene therapy in combination with chemotherapy aids in this respect. siRNA and miRNA alone or in combination with chemotherapy improve therapeutic response in tumor cells. Some natural substances, such as curcumin, quercetin, tocotrienol, parthenolide, naringin, and cyclosporin-A are also helpful in combating the drug resistance of cancer cells. This manuscript summarizes the mechanism of drug resistance and nanoparticle-based strategies used to combat it.

## 1. Introduction

Cancer is a leading cause of death worldwide. According to the WHO statistics 2019, cancer is the first or second leading cause of death in 113 countries [1]. As per the WHO statistics for 2020, 19.3 million new cases of cancer and 10.0 million deaths were reported worldwide from cancer [2]. There have been extensive efforts for the delivery of drugs to tumor sites, but the developed drug resistance has led to a reduced therapeutic effect [3]. Drug resistance may be intrinsic or acquired but leads to relapse in most cases [4]. The initial solution to drug resistance in cancer has been combination therapy that can help to overcome drug resistance. Combination therapies can be the co-delivery of genes (siRNA, miRNA) along with chemotherapeutic drugs or a combination of natural substances that alter the mechanism of resistance in the tumor [5]. The use of nanocarriers has been effective in the delivery of drugs and genes at specific tumor sites and in overcoming resistance. These include peptide-modified and bioresponsive nanoparticles functionalized with targeting agents to penetrate the tumor microenvironment and help to overcome tumor resistance [6]. Different types of nanocarriers have been employed to target tumors, such as lipid polymer hybrid nanoparticles, gold nanoparticles, mesoporous nanoparticle antibodies, functionalized nanoparticles, and biomimetic nanoparticles [7]. There remains a need for a detailed understanding of different factors that lead to drug resistance in tumors and for additional strategies to combat the resistance mechanism. Here, we summarize the mechanism and factors involved in multidrug resistance and the latest strategies adopted to combat this problem.

## 2. Mechanisms of Multidrug Resistance

Drug resistance in cancer is a common cause of the poor response to chemotherapy and is associated with 90% of the mortality in cancer patients. The main causes of multidrug resistance (MDR) in cancer include the efflux of drugs, increased DNA repair capacity, genetic factors, and the increased metabolism of xenobiotics (Figure 1). These mechanisms lead to a decrease in therapeutic efficacy and treatment responses [4,8].

### 2.1. Enhanced Efflux of Drugs

The enhanced efflux of chemotherapeutic agents resulting in lower intracellular drug concentration is one of the major causes of chemotherapeutic resistance [9]. Trans-membrane transporters that are responsible for drug efflux are mainly from the ABC transporter subfamily. There are 48 ABC genes classified into 7 subfamilies [10]. ATP-binding cassette (ABC) proteins such as P-glycoprotein (P-gp) are present on the surface of the cell membrane and are responsible for the absorption and excretion of a variety of chemical compounds. P-gp is highly overexpressed on the endothelial cell surface and leads to reduced drug penetration at the specific tumor site. P-gp can efflux a variety of anticancer agents, including anthracyclines, taxanes, and vinca alkaloids, to decrease the intracellular drug concentration [11]. A correlation between the overexpression of P-gp and increased resistance is particularly associated with paclitaxel, doxorubicin, vinblastine, etoposide, and olaparib [12,13]. In some types of hematological cancer, an initial low expression and then a dramatic increase in the level of P-gp (ABCB1) after chemotherapy has been observed [14]. ABCG2 is mainly involved in drug efflux in breast tumor resistance. It can transport both cationic and anionic drugs including chemotherapeutic agents and imatinib [15]. The efflux of drugs can also be mediated by extracellular vesicles (EVs), MRP1, and BCRP proteins. The functionality of efflux pump also depends on the composition and characteristics of the plasma membrane [16].

### 2.2. Genetic Factors

Certain genetic factors also contribute to chemotherapeutic resistance such as gene mutation, amplification, and epigenetic alterations.

#### 2.2.1. Gene Mutations

Gene mutations mostly occur in tumor cells and are one of the major causes of chemotherapeutic resistance and poor treatment response. The basis for the development of MDR in cancer cells is their aneuploidy nature [17]. During the process of mitosis, the reassortment of chromosomes can lead to the loss of drug-sensitive genes with the resultant loss of chemotherapeutic resistance. By contrast, normal cells that do not lose or gain a chromosome retain their drug sensitivity [18]. Mutation in TP53 in cancer cells is a well-known biomarker for tumorigenesis. TP53 is responsible for regulating the cell cycle and inducing apoptosis in cases of genotoxic stress during the process of replication. Mutation in TP53 genes also leads to the loss of tumor-suppressive activities [18]. Similarly, mutations in BCR-ABL genes associated with drug-binding regions lead to resistance in chronic myeloid leukemia (CML) [19]. Some studies support a correlation between the reactivation of BCR-ABL genes and remission of CML [20].

#### 2.2.2. Amplifications

Some chemotherapeutic agents such as methotrexate act by inhibiting key enzymes such as dihydrofolate reductase that control cell proliferation. A few types of cancer such as leukemia overcome this inhibition by enhancing the transcription of genes that encode the enzyme. This process is related to the synthesis of a specific region of a chromosome, and these amplified sequences are identified with double chromosomes. This results in the increased production of enzymes, and drug treatment strategies are not able to inhibit these enzymes, leading to reduced therapeutic activity [21]. The amplification of human epidermal growth factor receptor-2 (HER-2) leads to modification in a variety of genes and increases chemotherapeutic resistance to anti-HER-2 drugs [22].

#### 2.2.3. Epigenetic Alterations

Epigenetic alterations also contribute to chemotherapeutic resistance. Epigenetic alterations include histone modification, DNA methylation, non-coding RNA-related alterations, and chromatin remodeling [23]. DNA hypermethylation or hypomethylation is a potential factor contributing to cancer drug resistance. During tumorigenesis, the epigenome undergoes some alterations including the loss of DNA methylation, histone modification, and changes in miRNA expression [24,25]. The demethylation of the ABCB1 gene results in the decreased accumulation of chemotherapeutic agents in cancer cells. Epigenetic alterations may also affect DNA repair systems. Epigenetic changes related to miRNAs play a key role in the development of chemotherapeutic resistance in cancer cells. Studies have shown that miRNA affects the sensitivity of cancer cells to anticancer drugs through epigenetic changes [26].

### 2.3. Growth Factors

The results of various studies indicate an association between inflammation and cancer. Chronic inflammatory responses lead to tumor invasion. There is increased production of interleukin (IL-4, IL-6, and IL-8) and growth factors in MDR tumors compared with drug-sensitive cancers [27,28]. IL-6 can affect different biological processes such as cell growth and death by increasing the expression of the ABCB1 gene. There is convincing evidence of a correlation between the presence of IL-6 in tumor stroma and the MDR of gastric tumors [29]. The enhanced activity of protein kinase-C and extracellular matrix (ECM) in breast cancer is associated with increased resistance to chemotherapy [30,31].

### 2.4. Increased DNA Repair

Tumor cells also develop resistance to various chemotherapeutic drugs by increasing their ability to repair DNA damage. DNA repair endonuclease XPF and DNA excision repair protein ERCC-1 are essential for the repair of DNA damage induced by platinum-based agents [32]. There is a correlation between the increased expression of XPF and ERCC-1 and tumor resistance to cisplatin. Cisplatin and 5-fluorouracil both kill tumor cells by inducing DNA damage. Some genes involved in DNA repair mechanisms such as FEN1 and FANCG are reported to be overexpressed in 5-FU-resistant colon cancers [4].

### 2.5. Elevated Metabolism of Xenobiotics

Some enzymes responsible for drug metabolism can contribute to chemotherapeutic resistance. Drug-metabolizing enzymes are an essential part of phase-I and phase-II metabolism. Cytochromes P450 are key enzymes responsible for the metabolism of a variety of drugs. An increased expression of CYP1B1 has been observed in some cancer cells that modify the biotransformation of paclitaxel, docetaxel, and flutamide [33]. An increased expression of the CYP2A6 enzyme, which is involved in the metabolism of cyclophosphamide and 5-fluorouracil, has been observed in breast cancer tissues. The overexpression of CYP1B1 and CYP2A7 in different cancers is related to increased resistance to different chemotherapeutic agents [34].

## 3. Strategies to Overcome Multidrug Resistance

Multidrug resistance (MDR) results in a poor response to chemotherapy and is further increased after initial therapy because chemotherapy expands the resistant clones. Different strategies have been used to combat drug resistance, including restoring the oxidative stress of MDR cells, the downregulation of the genes causing resistance after identifying new targets for immunotherapy, and the use of chemosensitizers [35].

### 3.1. Use of siRNA to Combat Multidrug Resistance

The co-delivery of drugs and genes can help to overcome chemoresistance in tumor cells by either downregulating the genes causing resistance or through co-delivery by blocking two or more pathways that help in the death of tumor cells [36]. RNA interference (RNAi) has become significant for its role in specific gene silencing. Several studies support the downregulation of proteins causing resistance by RNAi [37].

Transcription factor Nrf2 is a key regulator of antioxidation and cryptoprotective systems. Nrf2 activation results in resistance to chemotherapy in many tumors including bladder cancer since tumor cells depend on its activity for survival. Therefore, Nrf2 can be a suitable target to overcome drug resistance. Different agents have been used to target NRf2, but they lack specificity; thus, the use of siRNA presents an attractive option to specifically target Nrf2 [36]. The use of a specific siRNA can help to overcome cisplatin resistance in bladder cancer. In one study, siRNA loaded onto carbosilane dendrimers (GCD) was used to combat cisplatin resistance in bladder cancer. The results showed that siNRf-GCD-based dendrimers increased sensitivity to cisplatin and improved the therapeutic response. The treatment of non-cancerous cells demonstrated a good safety profile of dendrimers with siNrf2 [38].

Focal adhesion kinase (FAK) is overexpressed in different types of cancer. The inhibition of FAK sensitizes the cells to chemotherapy. FAK also has a role in metastatic features, including invasion and migration. FAK siRNA may lead to the inhibition of FAK expression and help in combating resistance. In another study, hyaluronic-acid-modified PLGA nanoparticles were prepared for the co-delivery of paclitaxel and (FAK) siRNA to assess resistance in ovarian cancer. The prepared nanoparticles showed higher affinity toward CD44-positive cancer cells and resulted in significantly higher cytotoxicity and tumor growth inhibition compared with paclitaxel alone [39].

One of the important strategies used to overcome multidrug resistance is to decrease the overexpression of proteins that cause the efflux of drugs. P-gp and MRP1 proteins promote the efflux of drugs. In some cancers, such as prostate cancer, MDRP1 is more prevalent than P-gp [40]. The downregulation of drug resistance proteins such as MRP1 with siRNA can help to overcome drug resistance. The delivery of siRNA (siMRP1) using silicon-based nanoparticles provides an effective platform to combat doxorubicin resistance. The prepared nanoparticle consisting of porous silicon (psiNPs) showed excellent gene silencing and inhibition of MRP1 expression by 76%. Thus, the downregulation of MRP1 resulted in an increased cytotoxic effect of doxorubicin and an enhanced therapeutic potential [41]. Notch1 activates the AKT pathway, which leads to the epithelial-to-mesenchymal transition by the direct activation of MVP, thus leading to tumor resistance. Targeting Notch1 can help to overcome chemoresistance [42]. In a related study, mesoporous silica nanoparticles were prepared for the co-delivery of cisplatin and siRNA that specifically targets Notch1 (siNotch1). siNotch1 downregulated the genes causing resistance and led to the suppression of proliferation in hepatocellular carcinoma [43].

Aquaporins (AQs) transport water molecules and are considered to be overexpressed cancer-promoting factors in a variety of cancers, particularly AQP5 [44,45]. The use of AQP-5-siRNA led to increased apoptosis in breast cancer, suppressed metastasis, and enhanced sensitivity to chemotherapeutic drugs with a better therapeutic outcome [46]. Similarly, P-glycoprotein (P-gp) is responsible for the efflux of a variety of drugs. The use of siRNA was effective in silencing P-gp genes that helped to combat chemoresistance [47]. In a related study, polyamidoamine dendrimers were used to combat drug resistance to doxorubicin by the co-delivery of P-gp siRNA and doxorubicin. The results showed enhanced cytotoxicity against MCF-7 ADR cell lines and effectively downregulated P-gp to overcome resistance to doxorubicin [48]. In another study, P-gp siRNA and doxorubicin were delivered using multiresponsive nanoparticles. Doxorubicin was loaded onto the inner PLGA core, and P-gp siRNA was adsorbed onto the modified cationic PEI (C16-S-S-PEI) shell that is reduction-/photoresponsive designed for the co-delivery of siRNA and doxorubicin. The dual-responsive design allowed tumor-microenvironment-specific release and the downregulation of the P-gp expression. The sequential release of P-gp siRNA and doxorubicin was observed in vitro and in vivo, which leads to the effective downregulation of P-gp and increased the efficacy of doxorubicin [49].

A triblock polymer (PEG-PLL-PAsp (DIP)) was prepared for the co-delivery of doxorubicin and siRNA that target Bcl-2 genes. The PEG layer provides stability to a poly L-lysine block, which forms a complex with siRNA. The efficacy was evaluated on DOX-resistant cell lines. The results showed enhanced therapeutic effects in a hepatoma model [50]. In another study, core–shell nanoparticles were prepared for the co-delivery of Bcl-2 siRNA and doxorubicin to combat resistance. The increased efflux and antiapoptotic defense led to drug resistance in tumors. To overcome resistance, DOX was incorporated as a core and B-cell lymphoma-2 (Bcl-2) siRNA as a covering shell for synergistic effects. PEG and PEI were used to enhance the stability of the prepared nanoparticles. The siRNA effectively inhibited the Bcl-2 protein expression, which protects tumor cells from apoptosis. The study showed that the co-delivery of Bcl-2 siRNA and DOX significantly enhances the therapeutic effect of doxorubicin and helps to overcome drug resistance [51]. In another study, a multifunctional envelop nanodevice (MEND) was used for the co-delivery of epirubicin and Bcl-2 siRNA. An acid-sensitive ketal poly amino ester (KPAE) was conjugated with siBCL-2 and further coated with lipids containing epirubicin to form EPi/siBCL-2 nanoparticles. These nanoparticles have been shown to improve cytotoxicity on HepG2 cells and effectively downregulate the protein causing drug resistance. MENDs may prove to be promising delivery systems for the co-delivery of chemotherapeutic drugs and siRNA to combat drug resistance [52] (Table 1).

The overexpression of the enhancers of zeste homolog-2 (EZH2) favors tumor progression. The inhibition of EZH2 increases the apoptosis of cancer cells and the efficacy of cisplatin. EZH2 promotes nucleotide excision repair (NER) and contributes to cisplatin resistance. Silencing EZH2 by using specific siRNA triggers the apoptosis of cancer cells and increased sensitivity to cisplatin chemotherapy [62]. Methotrexate (MTX) is used to treat various types of cancers including lung cancer and lymphomas. The monotherapy of MTX leads to resistance and decreased efficacy. Studies have shown that survivin can facilitate MTX resistance. The downregulation of survivin can increase the therapeutic efficacy of MTX. The combination of survivin siRNA and MTX can help to overcome drug resistance. MTX was conjugated with linolenic acid (LA)-modified PEI. The micelles were obtained through the self-assembly of MTX-PEI-LA-PEG and complexed with siRNA at an N/P ratio of 16:1. The micelles were effectively taken up by HeLa cell lines, downregulated the survivin expression, and increased the therapeutic output [63]. The ABCC_3_ gene promotes the efflux of cisplatin, and silencing ABCC3 genes can help to overcome cisplatin resistance. The co-delivery of ABCC3-siRNA with the drug to the targeted cancer cell can combat resistance. Hybrid nanocarriers comprising PEG-PLA were prepared for the co-delivery of cisplatin and ABCC3-siRNA. The results showed that an optimized formulation resulted in significantly higher cytotoxicity on A549 cells compared with a formulation without siRNA. siRNA effectively knocked down the ABCC3 gene and improved the efficacy of prepared nanoparticles. The optimized formulation showed significant tumor regression in an A549-xenografted nude mouse model, which led to improved therapeutic effects [54].

The overexpression of the myeloid cell leukemia-1 factor (Mcl-1) has emerged as a promising target for pancreatic cancer. Gemcitabine (Gem) is a first-line agent for the treatment of pancreatic cancer but most cancers develop resistance to gemcitabine due to the higher expression of Mcl-1. Therefore, a combination of MCL-1 siRNA and gemcitabine may help to overcome resistance and improve therapeutic responsiveness. A lipid-based drug delivery system was employed for the co-delivery of siRNA and gemcitabine named LP-Gem-siMcl-1 (Figure 2). The siRNA in this system effectively downregulated the MCl-1 expression and significantly enhanced the cytotoxicity of gemcitabine [64]. In another study, graphene-oxide-based nanoparticles were used for the co-delivery of doxorubicin and anti-VEGF siRNA. The results showed a 46–52% decrease in the expression of VEGF protein and the targeted co-delivery of siRNA and doxorubicin. An in vivo study also showed a significant reduction in tumor growth. The co-delivery of anti-VEGF siRNA and doxorubicin via grapheme oxide nanoparticles could be a promising approach to combat drug resistance [65]. siRNAs are effective in the downregulation of genes causing resistance and help to improve the therapeutic response. These effects are specific to the target gene and effective in silencing the overexpressed factors and help to improve the therapeutic response to drug treatments.

### 3.2. Use of miRNA to Combat Drug Resistance

MicroRNAs (miRNAs) regulate various genes that are involved in a variety of pathological and biological processes leading to cancer. miRNAs are involved in chemoresistance by influencing genes that cause apoptosis and cell proliferation. The downregulation of miRNAs that influence the genes causing resistance can help to combat resistance [26]. 5-Fluorouracil (5-FU)-based chemotherapy is commonly used in the treatment of colorectal cancer (CRC). Drug resistance usually develops and leads to a poor therapeutic response. In a previous study, exosomes were prepared for the co-delivery of 5-FU and miR-21 inhibitor. The prepared exosomes loaded with 5-FU and miR-21i were delivered to the 5-FU-resistant colorectal cancer cell line HCT-166^5FR^. The in vivo studies in a mouse tumor model showed exosomes could efficiently deliver 5-FU and miRNA and downregulate the miR-21 expression in 5-FU-resistant tumors. A significant reduction in the tumor was observed. The combination effectively reversed drug resistance and showed better efficacy compared with the single treatment with 5-FU or miR-21i [66].

In another study, a quantum dot (QD)-based nanocarrier was prepared for the co-delivery of 5-FU and microRNA-34a for the treatment of colorectal cancer. QD nanoparticles were conjugated with β-cyclodextrin and modified with TCP1 peptide ligand for the co-delivery of 5-FU and miR-34a. The results revealed that 5FU and miR-34a were efficiently encapsulated into TCP1-CD-DQ nanoparticles and showed significant tumor suppression in vitro and in xenografted mouse models [67]. The survival rate of cancer was accompanied by a poor response to chemotherapy and drug resistance. Calcium carbonate nanoparticles enveloped with the cell membrane (CM) were prepared for the co-delivery of miR-451 and doxorubicin for the treatment of bladder cancer. The miR-451 downregulated the expression of P-gp and overcame DOX resistance. The in vitro studies in VIU-87/ADR cancer cell lines showed a significantly improved therapeutic response compared with miR-451 or DOX alone. This co-delivery of miR-451 and DOX using CCM nanoparticles provides a suitable approach for targeted delivery to cancer cells [68]. A number of studies have indicated a correlation between cancer stem cells (CSCs) and chemoresistance. The synergistic targeting of bulk tumor cells and CSCs can help to reverse resistance in prostate cancer. A dual-responsive (pH and glutathione) nanocarrier was prepared for the co-delivery of docetaxel and miR-34 activator for the treatment of taxane-resistant prostate cancer. Docetaxel-loaded P-Rub nanoparticles were prepared by incorporating docetaxel into a pH-sensitive polymer di-isopropylaminooethanol (DIPAE) and the glutathione-responsive pro-drug P-rubone (P-Rub) to prepare self-assembled micelles. The micelles targeted the tumor cell via the glutathione-sensitive release of docetaxel and rubone. The rubone upregulated the intracellular miR-34a, which then downregulated the protein causing resistance [69] (Table 2).

Triple-negative breast cancer (TNBC) is difficult to treat because of enhanced resistance. It has been demonstrated that hsa-miRNA-143-3p plays a key role in drug resistance in TNBC. The downregulation of hsa-miRNA-143-3p increases MDR, while its upregulation leads to increased sensitivity and better therapeutic outcomes. To study these systems, a breast cancer tumor mouse model was prepared using paclitaxel-resistant TNBC cells. The exogenous hsa-miRNA-143-3p was specifically delivered to tumor cells. The overexpression of hsa-miRNA-143-3p significantly decreased the drug resistance and led to a better therapeutic outcome [75]. An approach to combat resistance in TNBC involved the co-delivery of anti-miR-21 and docetaxel using self-assembled chitosan derivatives. The chitosomes loaded with anti-miR021 and docetaxel showed excellent entrapment efficiency and increased cellular transfection. The combination of anti-miT-21 and docetaxel showed enhanced cytotoxic effects on TNBC cells compared with DTX or anti-miR-21 alone and provides a suitable platform for the study of resistance in breast cancer [76].

In another study, multifunctional magnetic core–shell nanoparticles (MCNPs) were prepared, which consist of a zinc–iron oxide (ZnFe_2_O_4_) core and a mesoporous silica shell for the co-delivery of let-7a microRNA and doxorubicin to overcome chemoresistance. microRNA let-7a has the ability to downregulate efflux pumps such as ABCG2 and suppress DNA repair mechanisms, resulting in the sensitization of chemoresistant breast cancer cell lines. The prepared nanoparticles were functionalized with the tumor-targeting iRGD peptide for targeted delivery. In vivo delivery was monitored using magnetic resonance imaging (MRI). The let-71 miRNA was shown to inhibit tumor growth by targeting RAS and HMGA2. The co-delivery of miRNA and doxorubicin using MNCP was shown to be effective in overcoming drug resistance and increasing therapeutic efficacy [77] (Figure 3).

Mitochondrial elongation factor 4 (mtEF4) is highly expressed in tumors compared with normal tissues. As the mtEF4 knockout results in mitochondrial dysfunction and cell death, the downregulation of mtEF4 can be beneficial for tumor treatment. MicroRNA (miR-31) inhibits the expression of mtEF4 and leads to cancer cell apoptosis. The co-delivery of miR-31 and doxorubicin helps to overcome resistance and leads to better therapeutic outcomes. Mesoporous silica nanoparticles were functionalized with PEI and hyaluronic acid for the delivery of miR-31 and DOX. The results showed significantly higher cytotoxic effects compared with miR-31 or DOX alone [78].

Cellular Fas protein with interleukin-1β-converting enzyme inhibitory protein (c-FLIP) is overexpressed in various tumors and leads to chemotherapeutic resistance. In colorectal cancer, the overexpression of c-FLIP leads to inhibition of chemotherapy-induced cell death. Therefore, the co-delivery of a chemotherapeutic drug and suppression of c-FLIP can help to overcome resistance. MicroRNA-708 downregulates the c-FLIP levels and helps to overcome resistance. A delivery system consisting of titanium-dioxide-based mesoporous nanoparticles as a core and a polymeric shell consisting of LPP and PEG-b-PLD was prepared for the co-delivery of paclitaxel and miR-708. The prepared nanoparticles were suitable for the co-delivery of miR-708 and paclitaxel, provided pH-dependent release in the tumor microenvironment, and showed significantly higher cytotoxic effects against HCT-116 and DLD-1 cell lines and increased cell apoptosis. The in vivo results in a xenografted tumor model also showed a significant reduction in tumor size [79]. MicroRNA therapy has unique properties useful for the treatment of cancer because of its ability to downregulate and upregulate the genes that play crucial roles in cancer. miRNA can cause drug resistance by affecting genes, and anti-miRNA can help to overcome that resistance. The co-delivery of miRNA along with chemotherapeutic agents using various nanodelivery systems helps to overcome chemoresistance and improves therapeutic outcomes [80].

### 3.3. Use of Chemosensitizers and Natural Substances to Combat Resistance

**Cyclosporin-A** has been used to improve taxane bioavailability after oral administration through increased oral absorption, decreased elimination, and increased tumor-cell sensitivity to the effect of chemotherapy. In one study, docetaxel and cyclosporine-A were co-encapsulated in a self-nanoemulsifying drug delivery system (SNEDDS) to improve the therapeutic response. Cyclosporin-A also inhibited the P-gp and P450 enzymes and improved the bioavailability of docetaxel. These SNEDDS models showed enhanced bioavailability of docetaxel after oral administration and better therapeutic outcomes. These SNEDDS models improved antitumor efficacy compared with intravenous docetaxel [81]. Cyclosporin-A improved the oral bioavailability and efficacy of gefitinib (Gef). The co-delivery of cyclosporin-A and Gef used with PEG-PLA nanoparticles improved the bioavailability and therapeutic efficacy in Gef-resistant cancer cell lines by inactivating the STAT3/Bcl-2 signaling pathway. This combination helped to overcome drug resistance and significantly enhanced the antitumor efficacy [82].

**Curcumin** is a natural compound that inhibits the metastasis of cancer cells via cell cycle arrest, and its combination with chemotherapeutic drugs helps to overcome resistance [83]. The co-delivery of curcumin and methotrexate improves the therapeutic response to methotrexate-based chemotherapy. PLGA-based nanoparticles were prepared for the co-delivery of MTX and curcumin. The prepared nanoparticles showed enhanced cytotoxicity on glioma cell lines. Curcumin caused cell cycle arrest at the G2/M phase and complemented the effect of MTX that inhibited the S-phase, leading to enhanced apoptosis [84]. In another study, core–shell nanoparticles were prepared for the co-delivery of curcumin and paclitaxel to reverse resistance in ovarian cancer. PEI was conjugated with stearic acid to prepare the “core” and was modified with hyaluronic acid (HA) used as the shell. The prepared nanoparticles targeted the CD44 receptors on the ovarian tumor and exerted a synergistic effect on drug-resistant SKOV3 cell lines. The combination also inhibited P-gp, as evident by Western blotting analysis. The in vivo results also revealed a significant reduction in tumor size compared with paclitaxel or curcumin alone [85].

**Naringin** is an effective inhibitor of breast cancer resistance protein (BCRP) and helps to overcome resistance in BCRP-mediated multidrug resistance and is also used as a P-gp inhibitor. The combination of naringin with paclitaxel helped to improve anticancer activity. Polymer-based mixed micelles were prepared for the co-delivery of paclitaxel and naringin. The results showed better cytotoxicity on breast cancer cell lines and offered an effective approach to combat drug resistance [86]. Naringin also inhibits STAT-3, AKt, and Bcl-2 expression and plays a role in combating drug resistance. Treatment with naringin reduced the tumor size in mice-bearing tumors [87].

**Tocotrienols** are used along with chemotherapeutic agents to increase cytotoxicity and sensitize the tumor cells to chemotherapy. The co-delivery of gemcitabine and tocotrienols was performed using nanovesicles to increase the therapeutic effects of gemcitabine against pancreatic cancer cell lines. The prepared nanoformulation showed a nine-fold increase in cytotoxicity and significantly higher cellular uptake [88]. A nanoemulsion formulation was prepared for the co-delivery of tocotrienol and cisplatin along with caffeic acid for enhanced antitumor activity. The prepared formulation was evaluated using A549 and HEP G2 cell lines. The co-delivery system showed significantly enhanced cytotoxicity due to the increased late apoptosis and generation of ROS that cause cell cycle arrest. The combination synergistically improved therapeutic efficacy against cancer cells [89].

**Resveratrol** is a chemosensitizer that exerts an effect through the downregulation of NF-kβ and STAT pathways and upregulation of p53 and p21 [90]. Resveratrol can induce the apoptosis of cancer cells and inhibit metastasis via the upregulation of tumor-suppressive miRNAs [91]. A study was carried out to evaluate the effect of resveratrol on doxorubicin-resistant MCF-7 cancer cell lines by modulating miR-122-5p. The results showed that resveratrol induces chemosensitivity in doxorubicin-resistant cancer cells by modulating miR-122-5p, which plays a role in apoptosis and cell cycle arrest [92]. In another study, polymeric micelles were prepared that consisted of mPEG-PLDA for the co-delivery of docetaxel (DTX) and resveratrol (RES) to combat docetaxel resistance. The combination of DTX and RES at a ratio of 1:1 showed better cytotoxicity against MCF-7 cancer cell lines [93]. A hydrogel formulation was prepared using resveratrol microspheres and cisplatin for the treatment of liver cancer ascites. In vitro cytotoxicity was observed using H22 cell lines. The in vivo mouse model was prepared by inducing advanced hepatocellular carcinoma (HCC) with ascites, and the therapeutic potential of Pluronic F127 hydrogel was evaluated. The results revealed that combination with cisplatin resulted in the inhibition of ascites and prolonged survival. The augmentation of cisplatin activity further showed that hydrogels are safe, with low toxicity, and have potential for clinical applications for the treatment of HCC [94].

**Quercetin (QC)** is a flavonoid that shows promising ability to induce apoptosis, inhibit angiogenesis, and exert antiproliferative activity against cancer cell lines. QC also inhibits P-gp and MRP1, which cause the efflux of chemotherapeutic drugs [95]. The combination of QC and chemotherapeutic drugs helps to combat drug resistance. Quercetin and doxorubicin were co-delivered using mesoporous silica nanoparticles for the treatment of gastric cancer. The in vivo efficacy in mouse models revealed the significantly enhanced efficacy of the prepared nanoparticles [96]. Chitosan-based nanoparticles were prepared for the co-delivery of quercetin and paclitaxel to overcome paclitaxel resistance in lung cancer cells. The prepared nanoparticles showed suitable physicochemical properties and synergistic activity against A549 cell lines and helped combat resistance by inhibiting Akt and ERK phosphorylation, which is the main pathway of paclitaxel resistance [97]. Targeted polymeric nanocapsules were prepared for the co-delivery of docetaxel and quercetin for the treatment of prostate cancer. The active targeting was performed by conjugating luteinizing hormone-releasing hormone (LHRH) with PLGA using PEG as a spacer. Conjugation was confirmed using NMR and FTIR. The in vitro cytotoxicity was evaluated on PC-3 and LNCaP cancer cell lines. The results revealed the significantly higher cytotoxicity, cell uptake, and caspase-3 activity of these nanocapsules. In vivo mouse models also showed a significant reduction in tumor size, thus suggesting that this combination can provide a suitable approach for the treatment of pancreatic cancer [98].

**Parthenolide** is a natural sesquiterpene lactone and has anticancer activity. Many cancers develop resistance to chemotherapy by the activation of the nuclear factor kappa (NF-kβ) signaling pathway. Parthenolide (PT) has the ability to inhibit NF-kβ signaling, induce apoptosis, and kill cancer stem cells along with bulk cells [99]. Parthenolide in combination with cisplatin is effective in combating resistance in non-small-cell lung cancer (NSCLC) by inhibiting the PI3K/AKT signaling pathway. The combination of PT and cisplatin leads to apoptosis in A549 and PC9 cells. Western blotting analysis showed levels of PI3K, AKt, caspase-3, and Bcl-2 were downregulated after treatment with parthenolide. The combination of cisplatin and PT provides a suitable option to combat drug resistance in NSCLC [100]. Nanocrystals were prepared for the co-delivery of parthenolide and sorafenib for the treatment of hepatocellular carcinoma (HCC). The co-delivery of PT and sorafenib showed significantly higher cytotoxicity against HepG2 cells. Nanocrystals were also tested using a mouse tumor model. The results revealed an 81% inhibition rate compared with PT (48%) and sorafenib (58%). The combination of PT and sorafenib can be a suitable option for treatment of HCC [101]. In another study, nanodiscoidal phospholipid micelles were prepared for the co-delivery of paclitaxel and parthenolide for the treatment of lung cancer. The nanodiscoidal micelles exhibit spherical shapes, long-circulating characteristics, and good loading capacity of PT and PTX. The combination showed significantly enhanced cytotoxicity against A549 and MDA-MB-231 cell lines. The studies using an in vivo tumor model also showed a significant reduction in tumor size [102].

## 4. Conclusions

Tumor resistance is a serious concern for effective therapeutic response. In recent years, different strategies have been adopted for combating drug resistance in cancer. The main reasons for drug resistance are the efflux of drugs and genetic factors. Gene therapy (siRNA, mRNA) has been effective in overcoming resistance, but it needs effective non-viral vectors to produce a response at desired sites. Different types of siRNA downregulate specific genes that cause drug resistance such as si-pgp, si-Bcl-2, siRNA STAT3, and ABCC3. Similarly, messenger RNA includes miRNA-21i, mRNA 34a, and mRNA 181. The combination of drugs and genes is effective for combating drug resistance, and nanoparticle-based systems with different modifications are suitable delivery vehicles. These include the delivery of miRNA using cationic polymeric nanoparticles and lipid nanoparticles. Recently, ionizable lipids also received considerable attention for their effective gene delivery. Creating different modified libraries of lipids and preparing nanoparticles using these modified lipids could provide a more effective system for the combined delivery of drugs and genes. The drug resistance in tumors can be overcome using targeted nanomedicine-based strategies that deliver drugs and genes to a specific tumor microenvironment and helps in improving therapeutic response. Studies are underway for the bench-to-clinic translation of these drug-and-gene co-delivery systems.

## Figures and Tables

**Figure 1 cancers-14-04123-f001:**
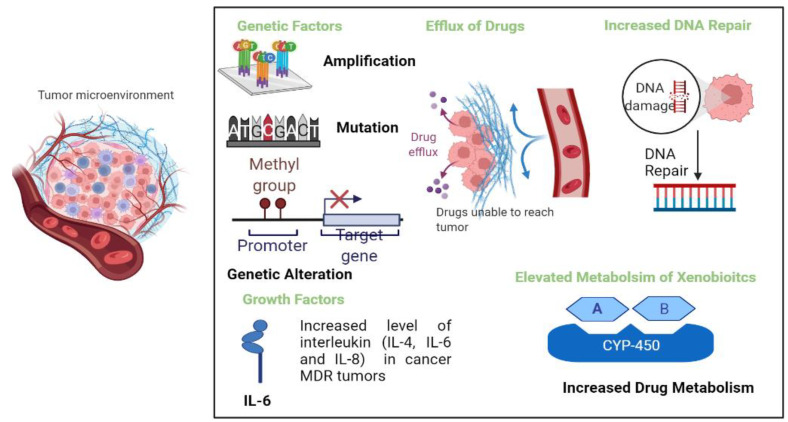
Different mechanisms of multidrug resistance in tumor cells, namely efflux of drugs, genetic factors, growth factors, elevated metabolism of xenobiotics, and increased DNA repair.

**Figure 2 cancers-14-04123-f002:**
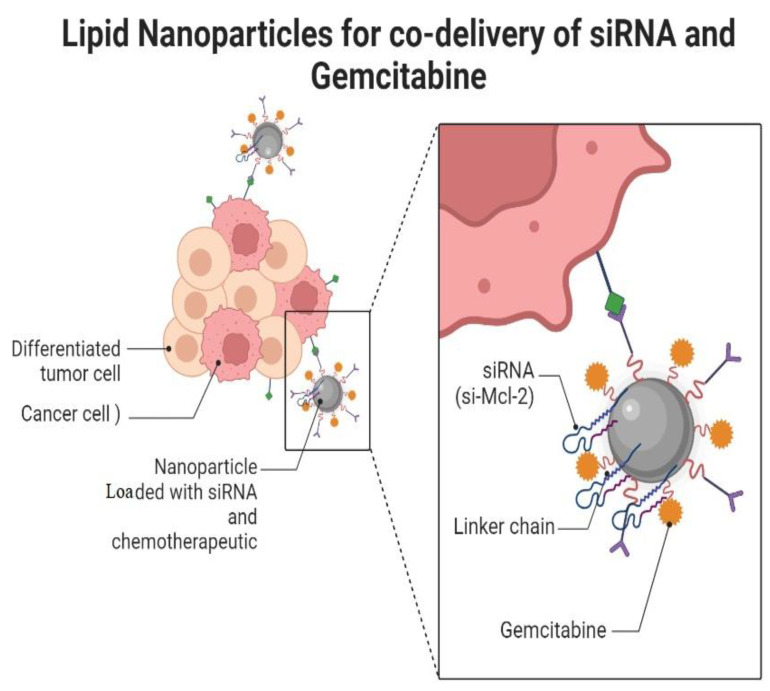
Nanoparticle-based delivery of siRNA and gemcitabine functionalized with targeting moiety, which delivers drug and Mcl-2 siRNA to cancer cells and helps to combat resistance.

**Figure 3 cancers-14-04123-f003:**
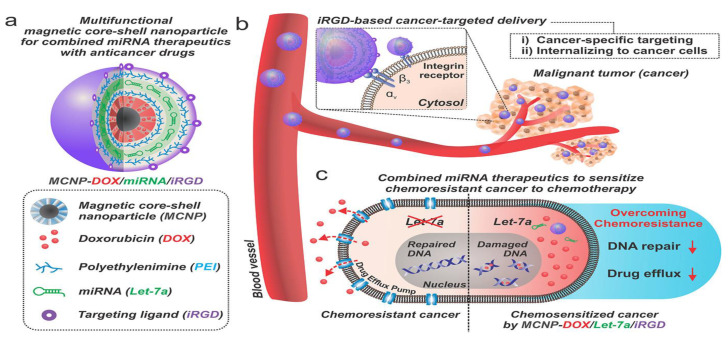
Multifunctional magnetic nanoparticles for co-delivery of miRNA-let7a and doxorubicin to improve therapeutic response and overcome drug resistance [77]. Copyright © 2018, American Chemical Society. (**a**) Design of the MCNP construct for simultaneous delivery of miRNA and anticancer drugs (e.g., DOX).; (**b**) MCNP construct is decorated with iRGD peptide, enabling the tumor-targeted delivery via α_V_β_3_ integrin-mediated uptake. (**c**) Co-delivery of miRNA (let-7a) sensitizes cancer cells to DOX therapy via inhibition of multiple chemoresistance-related genes, including those associated in drug efflux and DNA repair mechanisms.

**Table 1 cancers-14-04123-t001:** Co-delivery of chemotherapeutic drug and siRNA using different nanovehicles.

Name of Drug	siRNA	Nanovehicle	Cell Line or Animal Model	Reference
**Doxorubicin**	si-BCL-2 siRNA	(ATRA) double grafted N,N,N-trimethyl chitosan (TMC) nanoparticles	QGY-7703 cells H-22 tumor model	[53]
**Cisplatin**	ABCC3-siRNA	Hybrid nanocarriers (PEG-PLA)	A549 xenograft model of NSCLC	[54]
**Doxorubicin**	P-gp siRNA, Bcl-2 siRNA	Biodegradable boronic-acid-modified ε-polylysine	Breast cancer cell line (MCF-7/ADR) cells	[55]
**Salinomycin**	siRNA	Cholesterol-loaded chitosan nanoparticles (C-SAR)	Gastric carcinoma cells (SNU-668 and SGC-791	[56]
**Adriamycin**	siRNAs targeting MVP and BCL2	Multifunctional Carboxymethyl chitosan nanoparticle	Esophageal squamous cell carcinoma mice model	[57]
**Doxorubicin**	P-gp siRNA	GSH reduction- and photoresponsive polymeric nanoparticles	MCF/ADR cells	[49]
**Doxorubicin**	polo-like kinase I (plk1) siRNA	Polyethylenimine-modified ATRP-fabricated Polymeric Nanoparticles	MDA-MB-231 and HeLa cells EAT Tumor-bearing mice	[58]
**Methotrexate**	STAT3 siRNA	Chitosan-modified MSNs	MCF7 cells and breast cancer model	[59]
**Paclitaxel**	siRNA against HER2 (siHER2)	Targeted nanoparticle	Breast tumor and brain tumor	[60]
**Crizotinib (CRI)**	Bcl-xL siRNA	Cationic liposomes	MCF-7 cells and breast cancer model	[61]

**Table 2 cancers-14-04123-t002:** Nanomedicine-based system for co-delivery of miRNA and chemotherapeutic drugs.

Name of Drug	miRNA	Nanovehicle	Cell Line or Animal Model	Reference
**Paclitaxel**	miR122	Multivalent RNA nanoparticle	Hepatocellular carcinoma mice model	[70]
**Doxorubicin**	miR-21 inhibitor	Calcium phosphate-polymeric nanoparticle	MDA-MB-231 and A549 cells	[71]
**Doxorubicin**	miRNA-34a	Mixed nanosized polymeric micelles along with TAT peptide	HT1080 cells	[72]
**Doxorubicin**	miR159	Exosomes nanovehicle	MDA-MB-231 cells and TNBC breast cancer model	[73]
**melphalan**	miR-181a	Lipid nanoparticles	RB cells and retinoblastoma mice model	[74]

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
