# Peer review of "Recent Trends in Nanomedicine-Based Strategies to Overcome Multidrug Resistance in Tumors"

_cancers, 2022, doi:10.3390/cancers14174123_

Round 1

Reviewer 1 Report

In this review, Muzamil Khan and Torchilin review the role of nanomedicine in overcoming multidrug resistance. The topic is very interesting but some concerns need to be addressed. The comments are as follows:

·         In the abstract, mention some strategies based on nanoparticles to overcome drug resistance.

·         The introduction section is poorly presented. Refer to the latest update of cancer deaths according to the World Health Organization. Mention nanomedicine and its importance in drug/gene delivery in the introduction section. Refer to the latest papers published in the field of nanomedicine. (https://doi.org/10.1016/j.mtbio.2022.100208, https://doi.org/10.1016/j.semcancer.2022.02.006, https://doi.org/10.1016/j.addr.2021.114034, https://doi.org/10.1016/j.jconrel.2021.11.024)

·         The captions of the figures should be presented in more detail.

·         Different mechanisms of efflux of drugs are not well described. Some drugs are released through the channels in the membrane. The types of drug exit routes from the cell should be well described.

·         In general, the mechanisms of multidrug resistance should be presented in more detail and the research papers that focus on these mechanisms should be mentioned.

·         Contrary to the title, which emphasizes nanomedicine, it is very poorly presented in the strategy section. It is necessary to address the studies and achievements of overcoming multidrug resistance based on nanoparticles in more detail. It is also suggested to provide a table in this regard.

·         The conclusion section also needs to be revised.

Author Response

Reviewer 1

  1. As recommended by the Reviewer #1, in the Abstract, we have named some strategies based on nanoparticles to overcome drug resistance. 
  2. As recommended by the Reviewer #1 regarding adding WHO statistics and citing some papers in the Introduction, we have modified the Introduction and added WHO statistics about cancer and the relevant articles have been cited as well.
  3. As requested by the Reviewer #1 regarding the detailed captions,we have added such details to the captions of figures
  4. 4As requested by the Reviewer #1 regarding the detailed mechanism of efflux of drugs, the details on the drug efflux have been added.
  5. As suggested by the Reviewer #1 regarding adding more details on the mechanism of multidrug resistance,we are thankful to reviewer for nice suggestion, In this paper, we mainly focused on strategies to overcome resistance and brief details of mechanism are already provided to better understand these strategies. The more detailed analysis of MDR mechanisms would have required a whole separate paper. 
  6. As requested by the Reviewer #1 regarding the detailed strategies and related table,the details of strategies to overcome resistance are provided including siNRA-based strategies, miRNA-based strategies and chemo-sensitizers. Two tables are also provided in this regard

7. As suggested by the Reviewer #1, the conclusion has been modified

Reviewer 2 Report

In this interesting review the authors describe the various molecular mechanisms involved in the multidrug resistance (MDR) in cancer and the current advanced therapeutic strategies to combat it. The employment of a co-delivery therapy coupled with nanoparticles-based strategies offer significant advantages in increasing drug sensitivity and treatment response as well as a good safety profile. Overall, the manuscript describes interesting topics and the different sections are well exposed. The figures are representative. 

I believe the manuscript could be promising for a publication in cancers and could arouse wide interest for readers and researchers from various research fields.  

Very minor concerns:
-The English needs to be further improved
-various typing mistakes are present in the manuscript (e.g. Fig.2 "loaded" and not "loadded")
-Line 76: the authors stated that "Mutation in TP53 genes also leads to tumor-suppressive activities". Are the authors sure that this statement is right? TP53 gene is a tumor suppressor gene and mutations might lead to the loss of this suppressor function. 

Author Response

Reviewer 2

  1. As requested by the Reviewer #2,English has been checked by the native English speaker
  2. Responding the Reviewer #2’s  the typos have been corrected.
  3. Responding the Reviewer #2’s concern regarding TP53 gene mutation,we are thankful to the reviewer, the statement has been corrected.

Round 2

Reviewer 1 Report

Recommend to publish